# Cross-frequency coupling between slow harmonics via the real brainstem oscillators: An *in vivo* animal study

Yoshinori Kawai  *

Adati Institute for Brain Study (AIBS), Kawaguchi, Saitama, Japan

* ibs.stm.kwgt@gmail.com

## Abstract

Brain waves of discrete rhythms (gamma to delta frequency ranges) are ubiquitously recorded and interpreted with respect to probable corresponding specific functions. The most challenging idea of interpreting varied frequencies of brain waves has been postulated as a communication mechanism in which different neuronal assemblies use specific ranges of frequencies cooperatively. One promising candidate is cross-frequency coupling (CFC), in which some neuronal assemblies efficiently utilize the fastest gamma range brain waves as an information carrier (phase-amplitude CFC); however, phase-phase CFC via the slowest delta and theta waves has rarely been described to date. Moreover, CFC has rarely been reported in the animal brainstem including humans, which most likely utilizes the slowest waves (delta and theta ranges). Harmonic waves are characterized by the presence of a fundamental frequency with several overtones, multiples of the fundamental frequency. Rat brainstem waves seemed to consist of slow harmonics with different frequencies that could cooperatively produce a phase-phase CFC. Harmonic rhythms of different frequency ranges can cross-couple with each other to sustain robust and resilient consonance via real oscillators, notwithstanding any perturbations.

## Introduction

Rhythmic activity can be most conspicuously recognized in individuals by the ceaseless repetitive activities of respiration and heartbeats. Brain activity has also been characterized by rhythmic oscillations that underlie specific aspects of various neural functions, including sensation, locomotion, emotion, and cognition [1]. Rhythmic cardiorespiratory activity has been reviewed by its intimate and inseparable interrelationship with brain oscillatory activity [2–4]. Furthermore, it has been generally stressed that cardiorespiratory rhythms be generated based on pacemaker-like activity originating from particular brainstem neurons, rather than network activity, an oscillator hypothesis [5–7]. However, the body-brain interrelationship of rhythmicity has not been successfully quantized to the extent that fast Fourier transform (FFT)-based spectral methods involving time-resolved wavelet analyses have been applied to interpret any correlation between the dynamics of brain waves (delta to gamma frequencies) and bodily-recorded cardiorespiratory rhythms, except in our recent preliminary studies [8,

**Competing interests:** The authors have declared that no competing interests exist.

9]. To further explore the interpretation of brain-body rhythmic interrelation, stable simultaneous recordings of body and brain oscillatory activities from individuals with minimal surgical invasion under isoflurane-ketamine anesthesia were applied to rats in the present study for an FFT analysis of body-brain rhythm interrelationship. Ketamine is known to suppress breathing much less than most other available anesthetics [10–12], thus its adverse effects on the respiratory and circulatory systems are expected to be the least. To the best of our knowledge, similar procedures have not been applied to investigate cardiorespiratory rhythmic activities, except in our previous studies [8, 9]. Most previous investigations addressing peripheral and central cardiorespiratory rhythms used anesthetics with known respiratory suppression effects (such as pentobarbital or urethane). These investigations often involved artificial ventilation with a muscle relaxant and intravascular canulations for monitoring of vital signs, a severe surgical procedure that *per se* could cause any unexpected tremendous perturbation to naïve cardiorespiratory rhythmic activities [7, 13–15]. Our present recording methods, with minimal surgical invasion and controlled anesthesia with minimal respiratory suppression, enabled us to analyze the relationship between brain activity in the nucleus of the tractus solitarius (NTS) and peripheral cardiorespiratory rhythm recorded via a piezoelectric transducer (PZT), more accurately and stably than any previous studies.

Brain waves of discrete rhythms (gamma to delta frequency ranges) are ubiquitously recorded and discussed with respect to probable corresponding specific functions [1]. The recorded oscillations represent synchronization of electrical activity produced by the varied spatial sizes of brain modules that are activated simultaneously. Of these, it has been proposed that local activity within brain modules occurs at a high frequency (β and γ: 12–200 Hz) whereas long communication occurs at low frequencies (δ, θ and α: 0.5–12 Hz) [16]. Communication of various types between brain modules is likely to occur via nesting or cross-frequency coupling (CFC) [1, 17–21]. We addressed whether these criteria of communication across different frequencies of brain waves could be applied to the central and peripheral cardiorespiratory couplings and described several characteristics such as the harmonic waves seemingly unique to cardiorespiratory rhythms.

In this article, by focusing on the physiology and anatomy of a certain rodent brainstem region, the NTS, we demonstrated the characteristics of harmonic brain waves and their interactions, with a probable anatomical configuration of the responsible oscillator circuits. In addition, the possible responsible brainstem oscillators and their relation to large-scale brain regions were schematically postulated based on the anatomy of our own previous research and literature [22–24].

## Materials and methods

### Animal preparations and electrophysiological recordings

The experimental procedures and data analysis have been described previously [8, 9]. In brief, all surgical and experimental procedures were approved by the Institutional Committee for the Care and Use of Experimental Animals at the Jikei University School of Medicine in Japan, and were performed in accordance with the Guidelines for Proper Conduct of Animal Experiments by the Science Council of Japan. Electrophysiological recordings were performed using male Sprague–Dawley rats (weight range 280–310 g). The present study dealt with data obtained from 4 animals, and similar results were confirmed in other 3 animals. The animals were anesthetized with an intraperitoneal injection of ketamine (30 mg/kg) and xylazine (24 mg/kg) and placed in a stereotaxic instrument for recording. In most cases, 0.5% isoflurane was additionally administered through a nose mask to obtain a sufficient depth of anesthesia during recordings. Glass electrodes containing 2 M NaCl were used for *in vivo* extracellular

multi-unit activity (MUA) recordings. The resistance of the electrodes filled with this solution ranged from 1 MΩ to 5 MΩ. After making an incision in the atlanto-occipital dural membrane, an electrode tip was advanced vertically under a stereoscopic microscope with a motorized micromanipulator (IVM Single; Scientifica, East Sussex, UK) into the exposed left dorsal medulla at the level of the area postrema at a depth of 50–500 μm from the brain surface, aiming at the nucleus of the tractus solitarius (NTS) (Fig 7D; S2 and S3 Figs). Neuronal signals were recorded in the alternating current mode using a Multiclamp 700A (Axon Instruments, Union City, CA, USA). The amplified signals were analyzed offline using Spike2 (Cambridge Electronic Design Limited, Cambridge, UK) and IgorPro7 (WaveMetrics, OR, USA) software. For multiple local field potential (LFP) recordings, a 16-channel silicon probe (A1×16-Poly2s-5mm-50s-177-A16; NeuroNexus Technologies, Inc., Ann Arbor, MI) was used. The resistance of each electrode, specified by the manufacturer was between 0.96 and 1.17 MΩ. Each electrode "site" consisted of a circular platinum metal 15 μm in diameter, arranged by two 8-site-columns, and separated by 50 μm [25]. Electrical activities were amplified with the help of an amplifier (A-M Systems Model 3600 Amplifier; Carlsborg, WA, USA), sampled at 1–4 kHz, and stored for offline analysis. Cardiorespiratory activities were recorded noninvasively using a piezoelectric pulse transducer (PZT; MP100; AD Instruments, New South Wales, Australia). The PZT transformed the mechanical movement or vibration of the thorax (through touch on the sensor probe patch) into electrical signals that can be divided into heartbeat and respiration components [26].

## Data analysis

Neuronal signals recorded *in vivo* exhibited, to a highly variable degree, a mixture of single- or multi-unit spikes and LFPs, particularly when using standard glass electrodes, whereas signals recorded with a silicon probe mostly consisted of LFPs. For the 0–10 Hz phase (frequency range of cardiorespiratory fundamental rhythms) enhancement, neuronal signals were, in some cases, filtered offline with a low-pass type II Chebyshev filter (Spike2, low-filtered between D.C. and 100 Hz with an order of 2 and a ripple of 60) and expressed as low-pass (LP) filtered signals. Records of spontaneous signals (each 10–20 sec or 100 s epoch of data with ~1000 points) were sampled from each recording mode. Values are expressed as the mean ± standard error. Coherence and fast Fourier transform (FFT) power spectra were generated using IgorPro7. Continuous wavelet transform (CWT) and wavelet coherence (WCoh) using Morse wavelets (default wavelet function) were calculated to represent rainbow-colored scalograms using MATLAB (The MathWorks, Natick, MA, USA). The CWT and WCoh were expressed as time-resolved power and coherence spectra, respectively. A detailed explanation of each formula for the numerical analysis has been provided previously [8].

## Results

### Peripheral cardiorespiratory rhythms detected non-invasively as bodily-vibrations

Peripheral cardiorespiratory rhythms, as vital signs, were non-invasively monitored and recorded via a piezoelectric transducer (PZT) attached to the thorax (Fig 1). Ketamine anesthesia in combination with xylazine and isoflurane enabled stable long-duration central recordings from the brainstem with minimum mechanical perturbation in comparison to conventional pentobarbital or urethane anesthesia with severely invasive surgery, as concluded by our preliminary studies.

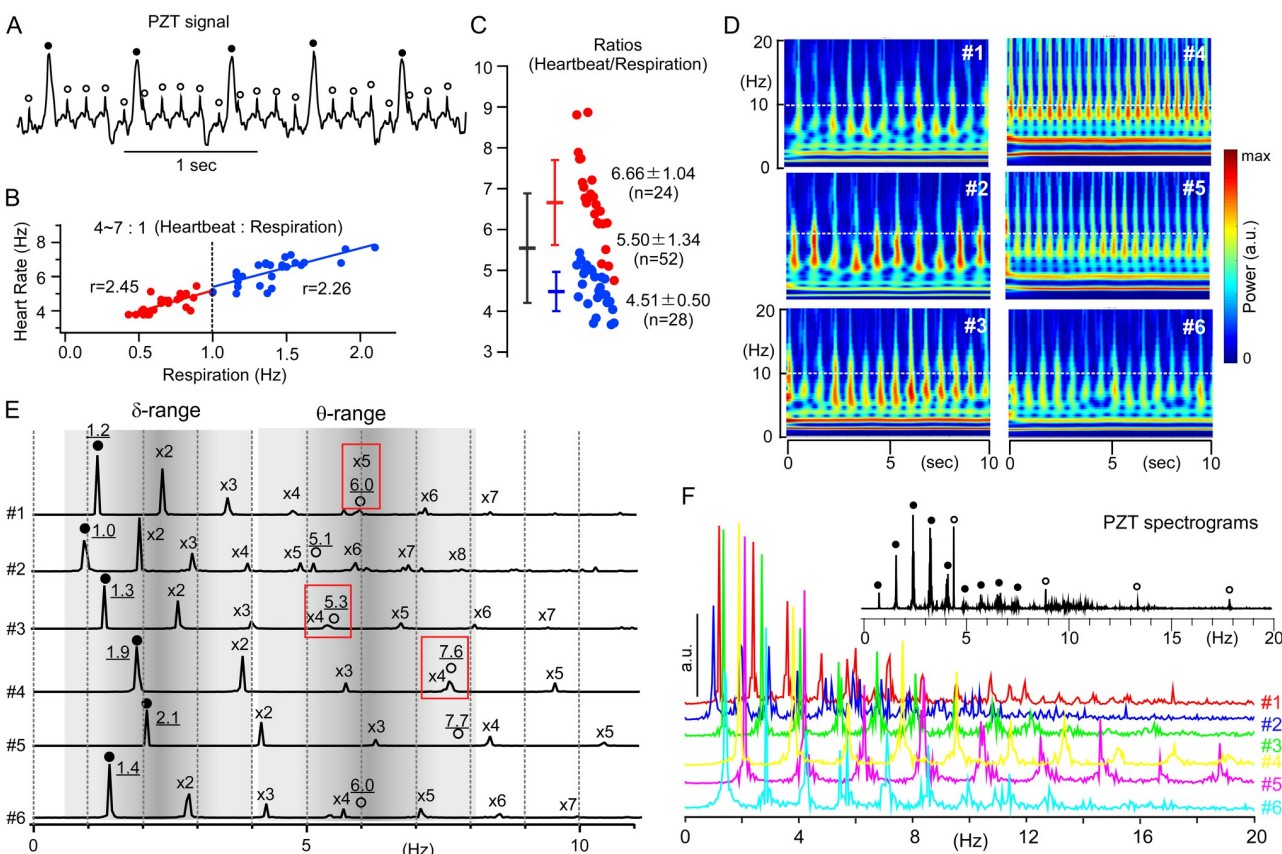

**Fig 1. Peripheral cardiorespiratory rhythms detected non-invasively as bodily-vibrations.** A, PZT signals had two kinds of periodical peaks (closed and open circles), which corresponded to respiration and heartbeat cycles, respectively. B, Respiration is ranged ~0.5–2 Hz (delta range), while heart rate is ~ 4–8 Hz (theta range). The slope of heartbeat/respiration (r) was ~2. Under stable anesthesia, respiration was more than 1 Hz (blue circles). C, The ratios of heartbeat / respiration (~4–8). Under more stable anesthesia (blue circles), the values are small with smaller variance, while deeper or unstable anesthesia often resulted in slower respiration and large ratios (heartbeat / respiration) with larger variance (red circles). The values of heartbeat and respiration calculated from 10 sec epochs of PZT recordings. D, FFT (fast Fourier transform) based continuous wavelet transform (CWT) of 6 examples (#1- #6) of 10 sec epochs of PZT signals from the same individual's recording of over ~2 hrs. Note regular continuous horizontal bands of strong power signals over delta frequency range (0.5–4 Hz) and vertical over theta range (4–8 Hz). E, FFT-based power spectra of #1–6 (10 sec epochs of recording) over delta (δ: ~0.5–4 Hz) and theta (θ: ~4–8 Hz) frequency ranges. Note fundamental (closed circles and underlined values) and overtone (x2–7) respiratory frequencies and fundamental cardiac frequencies (open circles and underlined values). Fundamental cardiac frequencies with red rectangles indicate a multiple relationship of cardiac / respiratory frequency ratios. Note that power of cardiac oscillations is much smaller than respiratory ones. F, Power spectra of peripheral cardiorespiratory rhythms (PZT) of 10-sec (colored) and 100-sec (black) epoch recordings. In shorter epochs of recordings (in color), oscillation powers seem to be conspicuous in delta and theta ranges as harmonic waves with no clear differentiations of respiration and heartbeat components. In contrast, oscillation powers analyzed with longer recordings emerge to differentiate into not only fundamental and overtone components but distinct respiration and heartbeat rhythms, indicating an existence of harmonic two different oscillators. The power peak of respiration oscillation resides in the second overtone (not fundamental) frequency. a.u.: arbitrary unit.

Respiratory and heartbeat frequencies from six 10-sec epoch recordings of PZT (#1–6) were calculated from the intervals of the respective cardiorespiratory signal peaks (closed and open circles: Fig 1A). The respiration and heartbeat were 1.08 ± 0.43 Hz (n = 52) and 5.46 ± 1.18 Hz (n = 52), respectively. The ratio of heartbeat to respiration rate was 5.50 ± 1.34 (n = 52) (Fig 1B and 1C). The profiles of the power spectra and time-resolved CWTs (Fig 1D–1F) revealed the existence of two distinct oscillators (delta and theta ranges). The CWTs (Fig 1D) indicated strong power signals of regular continuous horizontal bands over the delta frequency range (0.5–4 Hz) and intermittent vertical signals over the theta range (4–10 Hz). The perpendicular meshwork of robust signals over a 0–10 Hz frequency range suggested cross-

frequency coupling (CFC). The CFC was also represented by a longer epoch record (100-sec, Fig 1F). The longer-epoch record revealed two distinct oscillators (respiratory delta and cardiac theta, indicated by closed and open circles, respectively), both of which revealed harmonic characteristics of oscillation, in which there exists a fundamental frequency followed by several overtones, the multiples of each fundamental frequency (Fig 1F). Shorter epoch records showed varied fluctuating fundamental respiratory frequencies that ranged over the delta range (Fig 1D and 1E) with several overtones in comparison to the longer epoch record (Fig 1F). It should be noted that both the height of the signal peaks and the number of overtones were dominant for the respiratory component in the peripheral cardiorespiratory rhythm.

## Central rhythms recorded as multi-unit activities with glass electrodes

A long epoch (more than 1000 sec) recording using a glass micropipette placed in the dorsomedial medulla of the brainstem, the nucleus of the tractus solitarius (NTS), revealed multiunit activities (MUA) containing occasional polyphasic high-amplitude waves (Fig 2A) and conspicuous harmonic oscillations of MUA consisting of two distinct frequency ranges (closed circles in delta δ and open circles in theta θ gray rectangles) waves (Fig 2B). The fundamental frequencies of delta and theta oscillations were ~ 0.7 Hz and ~ 4.6 Hz, respectively. Both ranges of fundamental oscillations ensued several overtones, multiples of each fundamental frequency. In some cases, the power of theta oscillations was larger than that of delta oscillations (a, c, and d: Fig 2B). In the case of delta oscillations, the power peaks coincided with the non-fundamental overtone frequencies. Central harmonic oscillations shared several similar characteristics with peripheral cardiorespiratory oscillations. The most conspicuous difference between the central and peripheral oscillation patterns seemed to be the power balance between the delta and theta oscillations. The central rhythm in the NTS seemed to show more robust power in the theta oscillation than in the delta, whereas the peripheral rhythm appeared more in the delta as respiration than in the theta oscillations as heartbeats.

## Coherence of peripheral cardiorespiratory and central brainstem slow rhythms

Fig 3 shows two examples of shorter epoch (~10 sec) recordings for coherence analyses between peripheral cardiorespiratory activities recorded by PZT and central neuronal MUAs in the NTS recorded by glass electrodes. In comparison to power spectra by longer epoch (~100 sec) recordings, in which two distinct oscillator activities with fundamental frequencies and several overtones were distinguished (Fig 2), shorter epoch (~10 sec) recordings of central NTS revealed more fluctuating profiles of signal power dynamics of oscillations and their intimate relationship with the peripheral cardiorespiratory PZT rhythms (Fig 3). The multi-unit activities of the NTS neurons showed sporadic occurrence of polyphasic spikes of varied heights up to several millivolts from noise-level activity of hundreds of microvolts (Fig 3A1 and 3B1). The low-pass filtered activity of the NTS showed more robust signals than the NTS signals in the delta to theta frequency range (Fig 3A3 and 3B3). The profiles of the 10 sec-epoch (Fig 3A2 and 3B2) and time-resolved (Fig 3A3 and 3B3) power spectra of central (NTS, LP) and peripheral (PZT) signals showed, in general, similar qualitative patterns (Fig 3A2, 3A3, 3B2 and 3B3), which was confirmed quantitatively by a 10 sec-epoch (Fig 3A2 and 3B2) and time-resolved (Fig 3A4 and 3B4) coherence spectra between central and peripheral activities (Fig 3A2 and 3B2). The dynamics of power and coherence spectral profiles generally correlate between central and peripheral oscillations; however, several different features of their oscillation patterns, in terms of fluctuation, should be stressed, and the central oscillations

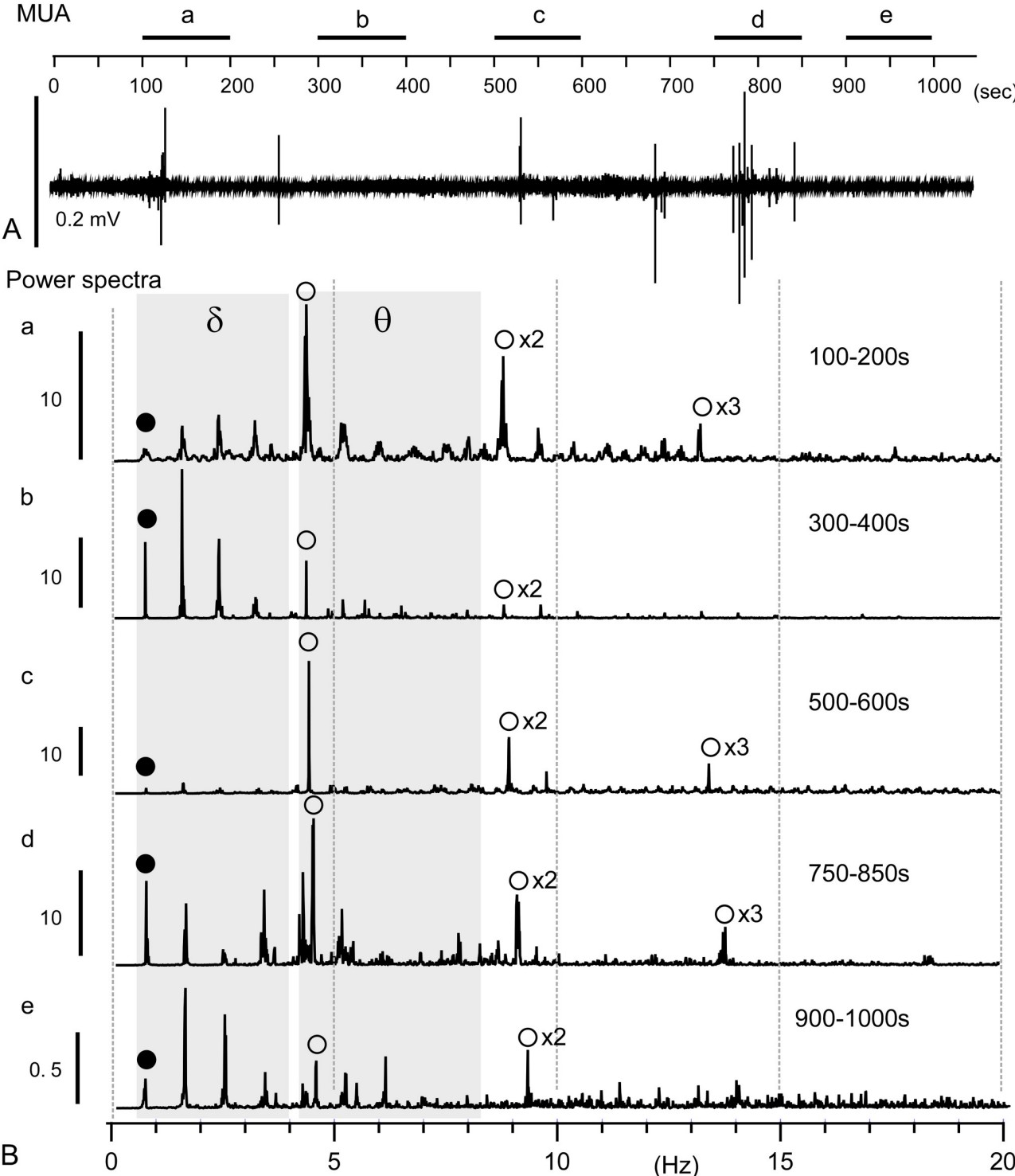

**Fig 2. Central delta and theta rhythms recorded as multi-unit activities with glass electrodes.** A, A long-epoch multi-unit activity (MUA) recording of over 1000 sec by a glass electrode. B, Power spectra of 100 sec epochs (a–e; also, in A). Closed circles indicate fundamental frequencies of respiratory oscillation, all of which reside in delta (δ) frequency range (grey rectangle: δ), followed by several overtones extending beyond a delta range. Open circles indicate fundamental cardiac oscillations (grey rectangle: θ) and their overtones (x2, x3 of the fundamental frequencies). Note that height peaks of respiratory power often reside in the overtones. Vertical bars in B indicate relative powers with respect to a common arbitrary unit.

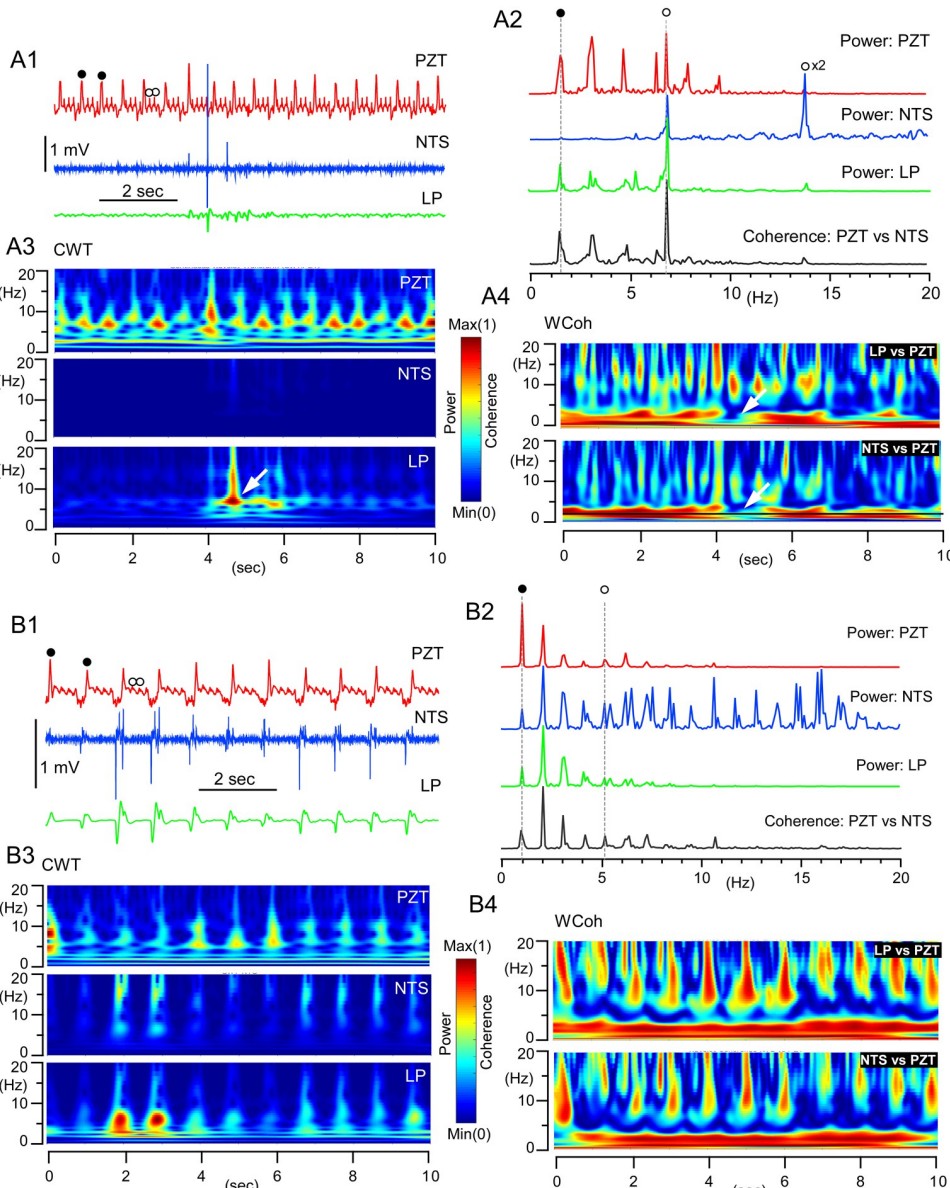

**Fig 3. Coherence of peripheral cardiorespiratory and central delta-theta rhythms.** A, B, Two examples of theta and delta temporarily dominant coherence of oscillations (~10 sec epoch recordings), respectively, recorded simultaneously from the NTS (central) and peripheral PZT. A1 and B1, Simultaneous records of PZT (in red), NTS (in blue) and the low-pass filtered (LP in green) signals. Scale bars of the amplitude height apply only to NTS signals, others were arbitrarily scaled. A2 and B2, Power spectra of PZT, NTS, LP signals of ~10 sec epochs of records. Bottom; Coherence spectra between PZT vs NTS. The heights of power and coherence spectra are arbitrary. Closed and open circles in PZT indicate respiratory and cardiac rhythms, respectively (A1, A2, B1, and B2). The first overtone of cardiac oscillation frequency is indicated by an open circle followed by x2 (A2). A3 and B3, Time-resolved power spectra (continuous wavelet transform: CWT) of PZT, NTS, and LP signals of ~10 sec records. A4 and B4, Time-resolved coherence spectra (wavelet coherence: WCoh) between LP (NTS) and PZT signals of ~10 sec records. Coherence was conspicuous as robust horizontal bands ranging delta frequency (0.5–4 Hz) and vertical ranging theta and over (5–20 Hz). Note the interruption of delta horizontal robust coherence indicated by arrows (A3 and A4).

showed more temporally varied activity, that is, a higher degree of fluctuation, especially in the theta frequency range, than the peripheral ones (indicated by white arrows in Fig 3A3 and 3A4). This temporal power fluctuation in the theta frequency range of the central oscillator gave rise to a temporary interruption of coherence in terms of robust continuous delta frequency activity between the central and peripheral oscillations, resulting in a consequent temporary break in cross-frequency coupling (CFC; Fig 3A3 and 3A4).

## A long record of simultaneous peripheral cardiorespiratory and central rhythmic activities beyond a period of cessation of oscillation

To examine the temporal dynamics of CFC more systematically, the relationship between two delta and theta oscillator activities and between peripheral and central oscillations was analyzed using a long record including a period of cessation of apparent oscillatory activity from an individual (Fig 4). When distinct cardiorespiratory (delta and theta) rhythms as PZT signals were evident (Fig 4A(a)–4A(c)), the central NTS signals seemed to exhibit similar oscillatory behaviors in terms of signal amplitude and timing (Fig 4A–4C). Fig 4B1 shows the time course of the simultaneous long records of PZT and NTS signals. The amplitudes (mV; mean ± standard deviation (SD) of the NTS signals (Fig 4B) were 0.28 ± 0.14 [n = 17, (a)], 0.86 ± 1.24 [n = 18 (b)], 0.07 ± 0.01 [n = 18 (c)], and 0.04 ± 0.01 [n = 16 (d)]. The amplitudes (V) of PZT signals were (Fig 4B) were 0.115 ± 0.012 (n = 18 (a)), 0.124 ± 0.027 [n = 18, (b)], 0.109 ± 0.009 [n = 18, (c)], and 0.075 ± 0.008 [n = 16, 8d)]. The respiration frequencies (Hz) of PZT (Fig 4B2) were 0.76 ± 0.09 [n = 27, (a)], 0.65 ± 0.15 [n = 21, (b)], 0.89 ± 0.04 [n = 28, (c)], 0.58 ± 0.05 [n = 26, (d)] and 1.11 ± 0.45 [n = 33, (e)]. The cardiac frequencies (Hz) of PZT (Fig 4B2) were 4.66 ± 0.21 [n = 36, (a)], 4.66 ± 0.29 [n = 35, (b)], 5.45 ± 0.22 [n = 36, (c)], and 5.12 ± 0.19 [n = 36, (d)]. The ratios of heartbeat to respiration were 6.13 (a), 7.17 (b), 6.12(c), and 8.82 (d).

When the apparent rhythmic PZT signals began to fade and almost disappeared during the long recording, the coherence dynamics between the PZT and NTS signals seemed obscure (Fig 4Cd1, 4Cd2, 4Ce–4Cg). A summation of PZT peak-triggered NTS and PZT activities (a closed circle in Fig 4Cd1) barely revealed small slow waves of NTS oscillation [Fig 4C(d2)] in accordance with the respiration rhythm of the PZT. This likely critical point of CFC was addressed more systematically by using time-resolved wavelet coherence (WCoh) analysis.

## Time-resolved coherence between peripheral cardiorespiratory and brainstem delta-theta rhythms during and beyond an apparent cessation of the oscillatory activities: The cross-frequency coupling break

Fig 5 shows a long record spanning beyond the apparent cessation of peripheral rhythmic activity (d) of time-resolved coherence between PZT and NTS signals, which represent peripheral and central rhythmic activities, respectively. Coherence between PZT and NTS signals was shown as intense signals horizontally in delta (δ) and perpendicularly in theta (θ) frequency ranges, along with an apparent bodily rhythmic PZT activity (Fig 5a–5c), however, the coherence was comparatively larger in either frequency range after the cessation of an apparent PZT activity (Fig 5f and 5g). During the cessation of apparent bodily rhythmic (PZT) activity (Fig 5d), the coherence showed a strange behavior that should be analyzed more critically.

Fig 6 shows the time-resolved coherence between the peripheral cardiorespiratory (PZT) and central delta-theta (NTS and LP) rhythms (D) during an apparent cessation of oscillatory activities (Figs 4 and 5) along with the original wave records (A), inspiration-triggered superimposition of signals (B), and FFT spectra of power and coherence (C). The bodily rhythms (PZT) consisting of respiratory and cardiac oscillatory activities (closed and open circles in Fig

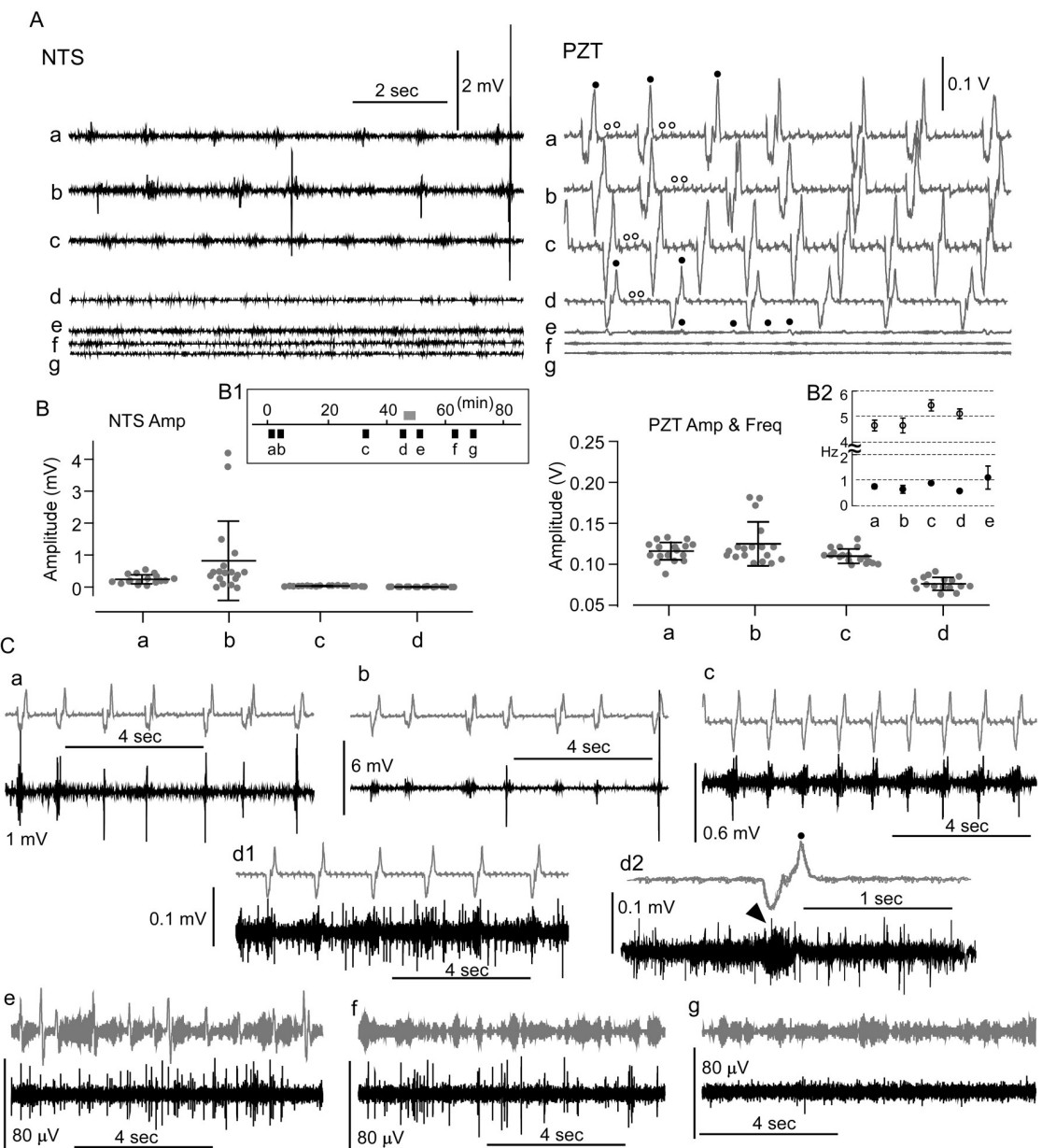

**Fig 4. A long record of simultaneous peripheral cardiorespiratory and central rhythmic activities beyond a cessation of apparent oscillation.** A, Simultaneous records of seven 10-sec epochs (a–g) over 80 min of central (NTS) and peripheral (PZT) rhythms beyond an apparent cessation of oscillatory activity. Apparent respiratory and cardiac rhythmic activities (indicated by solid and open circles) are evident in PZT signals. B, Amplitudes (Amp) of the central and peripheral rhythmic activities. The record epochs for the central (NTS) and peripheral (PZT) activities (B1), and frequencies (Freq) of the PZT rhythms (B2) are shown. C, Simultaneous records of PZT (in grey) and NTS (in black) activities from 10-sec epochs ((a)–(g) in B1). A grey bar in B1 indicates a period of an apparent cessation of the peripheral rhythmic activity (PZT). Note distinct signal [up to several mV in (a)–(c)] and noise [up to ~100 μV in (a)–(f)] activities. A summation (d2) of PZT peak triggered NTS and PZT activities (a closed circle in d1) reveals small slow waves of NTS oscillation (an arrowhead) in accordance with the respiration rhythm. An apparent cessation of both PZT and NTS activities is noted in (g). The amplitudes (mV; mean ± standard deviation (SD) of the NTS signals were 0.28 ± 0.14 [n = 17, (a)], 0.86 ± 1.24 [n = 18 (b)], 0.07 ± 0.01 [n = 18 (c)], and 0.04 ± 0.01 [n = 16 (d)]. The amplitudes (V) of PZT signals were (Fig 4B) were 0.115 ± 0.012 (n = 18 (a)), 0.124 ± 0.027 [n = 18, (b)], 0.109 ± 0.009 [n = 18, (c)], and 0.075 ± 0.008 [n = 16, 8d)]. The respiration frequencies (Hz) of PZT were 0.76 ± 0.09 [n = 27, (a)], 0.65 ± 0.15 [n = 21, (b)], 0.89 ± 0.04 [n = 28, (c)], 0.58 ± 0.05 [n = 26, (d)] and 1.11 ± 0.45 [n = 33, (e)]. The cardiac frequencies (Hz) of PZT were 4.66 ± 0.21 [n = 36, (a)], 4.66 ± 0.29 [n = 35, (b)], 5.45 ± 0.22 [n = 36, (c)], and 5.12 ± 0.19 [n = 36, (d)]. The ratios of heartbeat to respiration were 6.13 (a), 7.17 (b), 6.12 (c), and 8.82 (d).

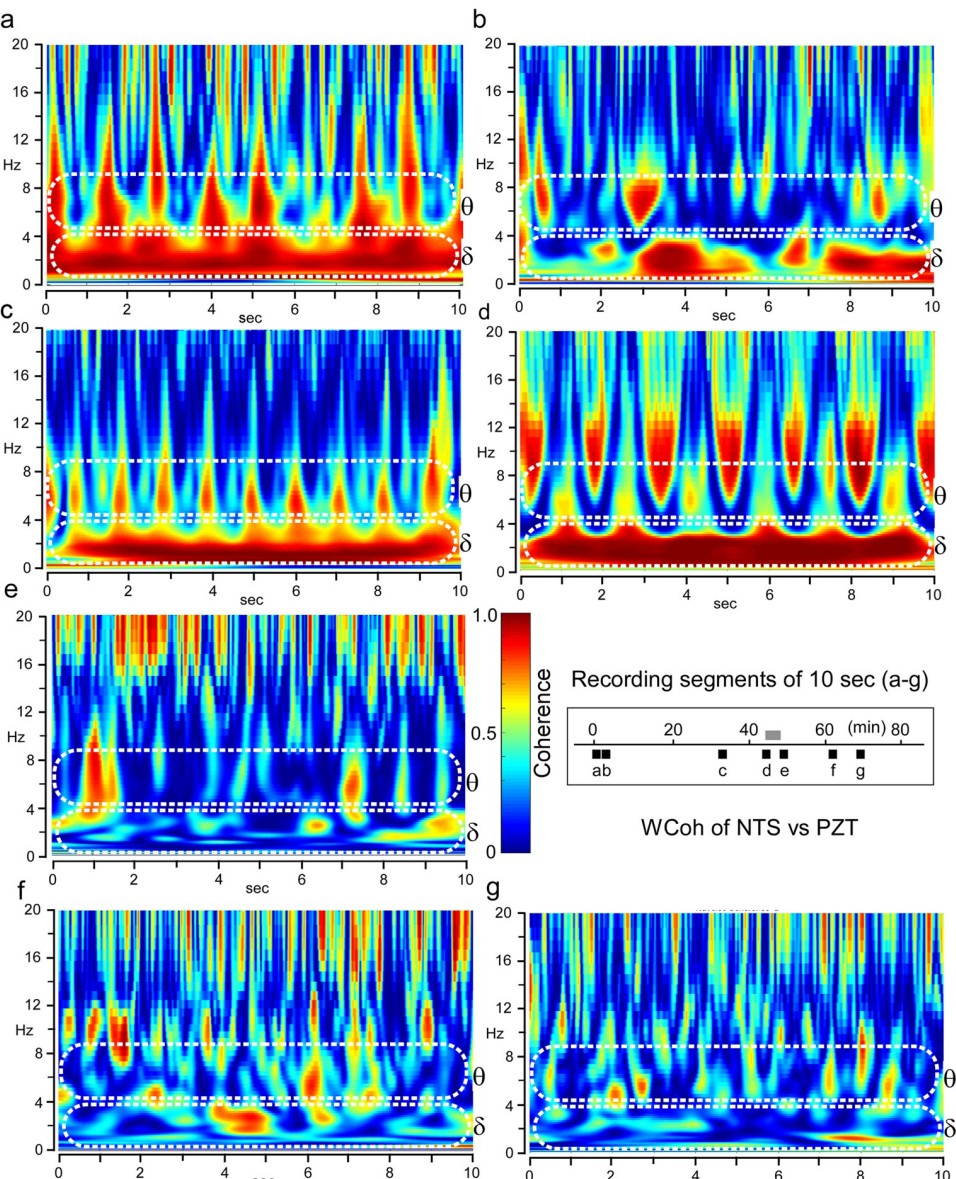

**Fig 5. Time-resolved coherence between peripheral cardiorespiratory and central delta-theta rhythms.** Time-resolved wavelet coherence (WCoh) analyses between PZT and NTS oscillations. Time series recording scheme of 7 epochs (each 10 sec duration; black bars designated by (a)–(g)) is shown with a grey bar indicating a period of an apparent cessation of the peripheral rhythmic activity. In each WCoh scalogram, two representative frequency range zones (delta δ and theta θ) are indicated by dotted white lines. Time-resolved coherence spectra (WCoh) between NTS and PZT signals reveal coherence is conspicuous as robust horizontal bands ranging delta frequency (0.5–4 Hz; δ) and vertical ranging theta and over (4–10 Hz; θ) in (a)–(d). Note an interruption of delta horizontal robust coherence signals in (b) and a complete discontinuation of robust coherence signals between delta and theta range in (d). With minor peripheral and central activities of a noise level, robust signals of delta and theta range and their CFC coherence almost disappear as shown in (f)–(g).

6A) that ranged in delta (δ) and theta (θ) frequencies, respectively (Fig 6C). It should be noted that the theta rhythm was quite small in the peripheral oscillation (PZT) and prominent in the central oscillation (NTS and LP). Second, and more importantly, delta rhythms in both peripheral and central oscillations lacked the characteristics of apparent harmonics that are

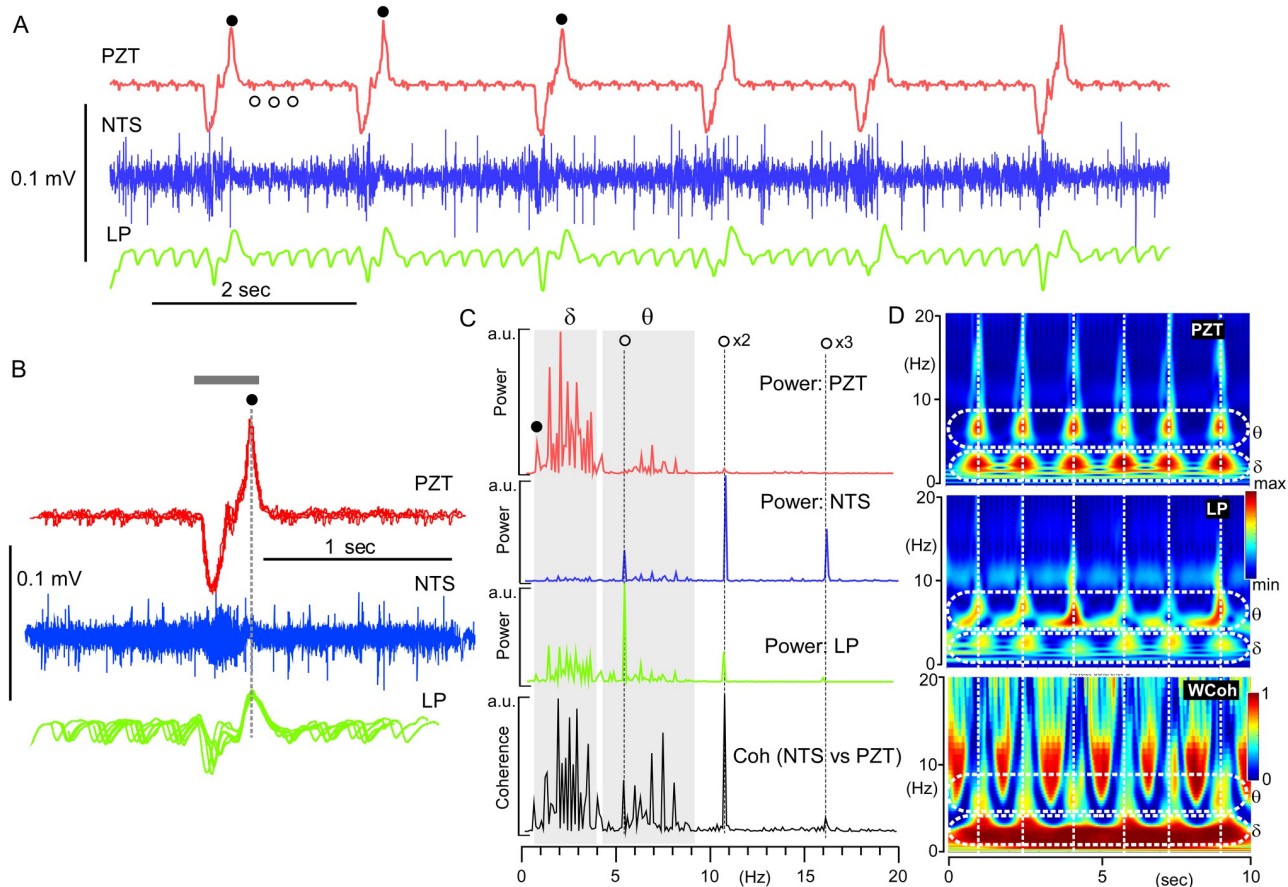

**Fig 6. Time-resolved coherence between peripheral cardiorespiratory and central delta-theta rhythms during an apparent cessation of oscillatory activities.** A, Simultaneous records of peripheral and central oscillations (a 10-sec epoch PZT in red and NTS in blue) just before an apparent cessation of bodily oscillatory activity (Fig 4Ae, 4Cd1 and 4Cd2), with a digitally low-pass filtered signal (LP in green) of the NTS signal. Amplitudes of PZT and LP signals are arbitrary. Apparent respiratory and cardiac rhythmic activities are indicated by solid and open circles in PZT signals, respectively. The frequencies of respiratory (closed circles) and cardiac (open circles) rhythms are 0.58 ± 0.05 Hz (n = 26) and 5.12 ± 0.19 Hz (n = 36), respectively. B, The PZT peak (indicated by a closed circle) triggered signals of NTS and LP. A grey bar indicates a period of inspiratory phase of respiratory cycles. C, Power spectra (Power) of PZT, NTS, and LP signals, and a coherence spectrum (Coh) between NTS and PZT signals. Note that the cardiac or theta (θ) rhythms assume harmonics with fundamental frequency (indicated by an open circle) followed by overtones (indicated by x2 and x3), while respiratory or delta (δ) oscillations lack apparent harmonic components. D, Time-resolved power (continuous wavelet transform: indicated by PZT and LP) and coherent (wavelet coherence: WCoh) analyses of PZT and LP signals. Note that frequency-coupling between delta (δδ) and theta (θ) oscillations breaks irrespective apparent persistent strong signals of each frequency (WCoh).

persistently recognized under stable anesthesia. As a result, the time-resolved coherence by wavelet coherence analysis (WCoh) behaved quite differently compared to that under stable anesthesia (Fig 5a and 5c), that is, robust horizontal and perpendicular signals representing delta and theta range oscillations, respectively, leave some trace of cooperative interactions in the scalograms (Fig 6D).

## Local field potentials of central delta-theta rhythms and the cross-frequency coupling

Fig 7 shows a long (~1 hr) record (A) of local field potentials (LFPs) using a multi-site silicon electrode that can simultaneously record neuronal activities from up to 16 separate sites (each 50 μm apart, D). LFPs contained sporadic episodes of high-amplitude signals up to several

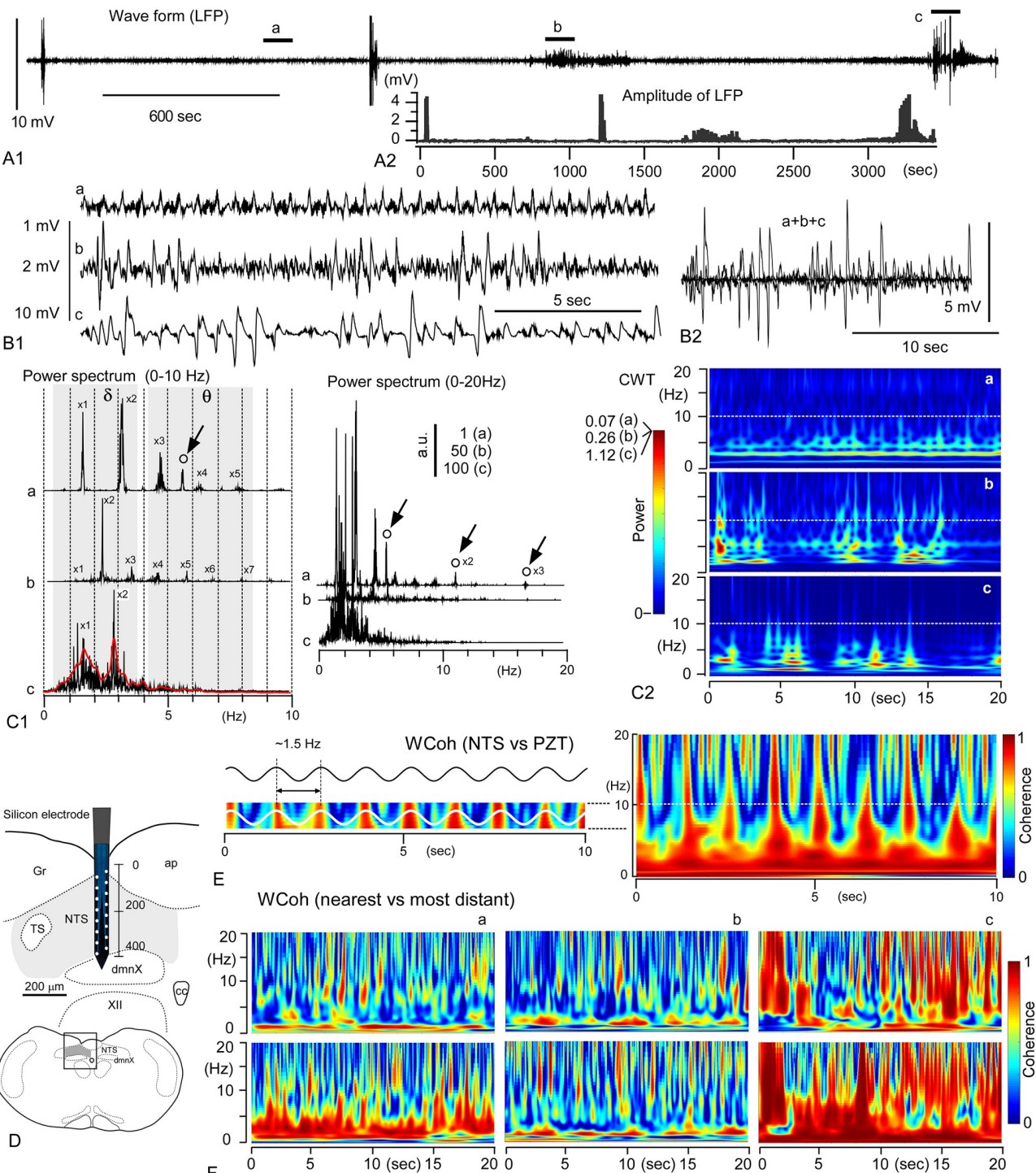

**Fig 7. Local field potentials of central delta-theta rhythms and the cross-frequency coupling.** A1, A long (~1 hr) record of local field potentials (LFPs) from a certain site of a simultaneous multi-site recording silicon electrode. A2, An amplitude of LFPs reaches several millivolts (mV) from the noise level activity of hundred μV. B, Three epochs of the records ((a)–(c) in A1) are arranged in tandem (with an arbitrary amplitude scale, B1) or in superimposition (B2). C1, Fast-Fourier transform power spectra of records (a)–(c). Delta (δ) rhythms with fundamental frequencies (x1) and overtones (x2, x3, x4. . .) are noted in (a)–(c). Theta (θ) rhythms are indicated by open circles (arrows, fundamental and overtones (x2, x3)) in records (a), but unnoticeable in (b) or (c). a.u., arbitrary unit with relative amplifications (1, 50, 100). A red line in (c) is an averaged spectrum. C2, Time-resolved power spectra (continuous wavelet transform, CWT) of records (a)–(c). Note moderate signals aligned horizontally and perpendicularly in delta and theta frequency ranges, respectively. Relative power scale (a)-(c) of arbitrary unit is applied to this set of CWT scalograms. D, A multiple-site silicon electrode for LFP recordings inserted in the NTS, the nucleus of the tractus solitarius in the dorsomedial medulla oblongata. ap, area postrema; cc,

central canal; dmnX, dorsal motor nucleus of the vagus; Gr, gracilis nucleus; TS, the tractus solitarius; XII, hypoglossal nerve. A scale indicates a distance in μm. E, Time-resolved coherence spectrum (wavelet coherence, WCoh) between peripheral cardiorespiratory rhythms (PZT) and central delta-theta rhythms of LPFs shows robust coherence signals aligned horizontally in the delta (δ) frequency range, and perpendicularly in the theta (θ) range. The robust theta signals behave as a sinusoidal oscillation of ~ 1.5 Hz that corresponds with delta rhythm frequency. F, Time-resolved of coherence (WCoh) of LFPs ((a)–(c)) recorded simultaneously between two separate sites according to differences in their distance (nearest 50 μm vs most distant 350 μm) and signal amplitude (x1 ~ x100, B1(a)–(c)). Robust coherence is confirmed consistently between different recording sites [(a)–(c)] in the delta frequency range irrespective of signal amplitude, while the distribution of robust coherence varies to a large extent according to the distance of recording sites or signal amplitude. For example, frequency ranges of robust coherence far beyond the delta are recognized when the signal amplitude increases even between distant sites (c). Note that frequency ranges of robust coherence are restricted in the delta and theta range when signal amplitudes are relatively small, and that as signal amplitudes become larger frequency ranges of robust coherence become rather far wider (c).

millivolts in persistent activities of less than ~100 μV in amplitude (Fig 7A and 7B). The power spectra of three records of 20 sec [(a)–(c)] confirmed a harmonic wave feature of central LFPs, in which the fundamental frequencies in the delta [indicated by x1, (a)–(c)] and theta (an open circle, (a)) ranges and their overtones (indicated by x2–x7) are recognized in Fig 7C1. Fig 7C2 shows the time-resolved power spectra (continuous wavelet transform, CWT) of each episode of LFPs (a)–(c), in which moderate signals were aligned horizontally and perpendicularly in delta and theta frequency ranges, respectively. The scalograms of LFPs recorded in the NTS were similar to those of peripheral cardiorespiratory PZT signals (Fig 1). This was confirmed by wavelet coherence (WCoh) analyses between the NTS LFPs and peripheral PZT signals (Fig 7E), indicating the existence of cross-frequency coupling between delta and theta range oscillations cooperatively synchronized between bodily and brain waves.

### Slow harmonics of peripheral and central oscillations, and their cooperative interaction through phase adaptions

Fig 8 shows a summary of the typical power spectral structures and dynamics of the slow harmonics generated by delta and theta oscillators. The slow harmonics were schematized according to records by three different methods: peripheral PZT, central NTS multi-unit activity (MUA), and local field potential (LFP; Fig 8A). In PZT, the powers of respiratory harmonics ranged in delta frequency (δ) were predominant with the maximal fundamental frequency, whereas in the NTS MUA, theta harmonics are more robust compared to delta oscillations, in which the first overtone power was greater than the fundamental frequency. In the NTS LFP, central oscillations were stochastically increased to a maximum degree (up to ~100 times in power) in the delta frequency range; therefore, the powers of the theta range oscillations appeared comparatively minimal. The typical structure and dynamics of power spectra were determined based on relatively longer epoch (~ 100 sec) records (Figs 1F and 2), while a shorter epoch (~10 sec) of records revealed tentatively more complex and stochastic behaviors in terms of cooperative dynamics of delta (δ) and theta (θ) harmonics (Fig 8B). In contrast to longer records, shorter records often reveal that the fundamental frequency of theta oscillations (in red) coincides with one of the overtones of delta wave frequencies (in blue). In this case, several combinatorial whole-number relationships were observed between the theta and delta oscillation frequencies (3:1–6:1 in Fig 8B).

## Discussion

### Cross-frequency coupling of slow harmonics via brainstem oscillator activities: Cardiorespiratory rhythm

Rhythmic oscillations of cardiorespiratory frequency range (0.5–10 Hz), corresponding to delta and theta brain ones, were consistently observed peripherally as bodily cardiorespiratory

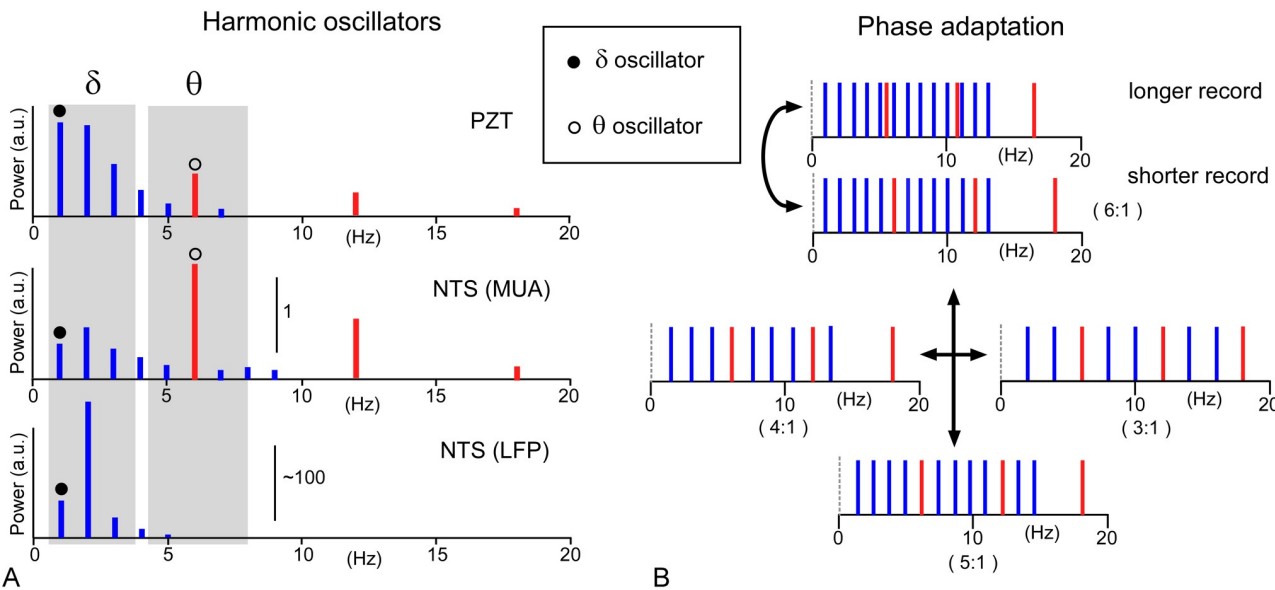

**Fig 8. Slow harmonics of peripheral and central oscillations, and their cooperative interaction through phase adaptions.** A, A summarized scheme illustrating power spectra of the harmonics generated by delta and theta oscillators. The harmonics are characterized by a presence of unique fundamental frequencies (closed and open circles in gray-shaded delta (δ) and theta (θ) frequency ranges) and their several overtones, multiples of each fundamental frequency (delta overtones in blue and theta ones in red). Robust powers of cardiorespiratory (PZT) and central (NTS) harmonics are seen with fairly specific patterns in delta (δ) and theta (θ) frequency ranges, respectively. Delta power is largest in the fundamental frequency (a closed circle in PZT), while theta power in the fundamental frequency (an open circle in NTS (MUA)). Note the largest power is often seen in the first overtone, not fundamental frequency in a case of LFP [NTS (LFP)]. Power scales apply to NTS MUA and LFP in arbitrary unit (a.u.). LFP, local field potential; MUA, multi-unit activity, PZT, piezoelectric transducer activity. B, The two oscillators (delta and theta, in blue and red, respectively) often stochastically form a relationship in which one delta overtone frequency coincides with the theta fundamental one (shorter record). Resonant harmonics ensue. The cooperative phase adaptation between the two oscillators could coordinate frequency of the harmonics (indicated by a double-headed curved arrow) to certain combinations of the multiple relationship (3:1 ~ 6:1) between the delta and theta fundamental frequencies.

rhythms and centrally as brainstem oscillatory electric activity with apparent cooperative dynamics (Figs 1–5). This has often been postulated, but not experimentally validated, probably due to inadequate methodological animal conditions (as discussed in the introduction section). In our study, two sets of oscillations were confirmed as respiratory (0.5–4 Hz) and cardiac (4–8 Hz) body rhythms and as delta and theta neuronal electrical activities in the brainstem. The peripheral and central oscillations or the two sets of slow rhythms both exhibited a feature of harmonics, that is, waves with particular fundamental frequencies followed by several overtones, while exhibiting cooperative interactions in terms of coherence and power with each other (Figs 7 and 8).

In general, an increase in the spatial coherence of brain activity results in an increase in the power of activity, as shown by WCoh (Fig 7, [8]). Thus, the coherence and power of neural activity wax and wane in concert within a frequency range of 0–15 Hz (delta δ and theta θ brain wave range) in the vagal complex (Fig 7). The fluctuation of brain wave power and coherence can be incessantly coordinated by the cross-frequency coupling (CFC) mechanism of cardiorespiratory frequency ranges (delta and theta ranges). In addition to the frequencies of slow wave use, the present study was the first to clarify that brainstem delta and theta waves exhibit conspicuous features as harmonics and interact with each other to coordinate rhythmicity suitable for probable varied specific functions.

It seems that by utilizing this rhythmic fluctuation of brain activity, specific clusters of neurons could communicate with others to perform functions such as respiratory and cardiac

regulation, interoception [27–29], hormone regulation for an adaptation to incessant change in environments. It is probable that such varied and different specific functions would mobilize functional networks to efficiently perform the corresponding required tasks. The functional networks thus tentatively and stochastically mobilized are generated through dynamic changes in brain wave power and coherence via a CFC mechanism [1, 30, 31]. Dynamic functional networks can behave differently, but are based on anatomically structured networks [32–35].

The spatiotemporal mobilization of specific neuronal clusters in the brainstem can accomplish tasks for each functional requirement via emergent oscillators. The coupling of delta- and theta-range brain oscillators can be best visualized by time-resolved spectrograms of coherence (wavelet coherence analysis: WCoh) or power (continuous wavelet transform analysis: CWT), as shown in the present study. The delta range oscillations of both brain waves (as NTS signals) and body (thorax) vibration (as PZT signals) would represent thin linear *horizontal* bands of highest values, while those of the theta range show distinct separate *vertical* thick bands or dots (Figs 1 and 3). As typically seen in WCoh spectrograms, they consist of horizontally continuous thick band(s) of high coherence ranging delta frequency and vertical bands of high coherence ranging theta frequency interspaced by delta-range frequency. They occasionally connect where coherence between brain waves and body vibration would grow large enough to give rise to large-scale oscillatory synchrony.

The resonant waves (harmonics) consist of a specific fundamental frequency and overtone. All the overtones or harmonics have a particular frequency, which is an integer multiple of the fundamental frequency. Cardiorespiratory oscillations possess two distinct fundamental frequency systems: delta and theta. Respiratory delta-range oscillation plays the role of a power supplier, whose efficiency is modulated by cardiac theta range oscillation via a mechanism of CFC, the origins of which should be explored in the brainstem delta and theta oscillators.

We speculate that stochastic CFC would temporarily utilize, and consequently constitute, the most reliable functional network suitable for responsible tasks based on the involved anatomical networks. This phenomenon of CFC can guarantee a reproducible and reliable task in response to a certain stereotypical input stimulus. We would like to propose a hypothesis that a certain functional network temporarily generated via a CFC mechanism could underlie reliably reproducible management of stereotypical tasks, and the functional networks with temporary and stochastic features could thus be created resiliently in response to an environmental change based on the anatomical networks of the solid framework.

Stochastic phase adaptation in concert with signal amplitude amplification of brainstem oscillations could provide a reliable and reproducible signal processing strategy in terms of the quality and quantity of information transportation [9]. The cooperative interaction of signal phase and amplitude would be basically stochastic, but may provide a resilient and robust adaption capability for complex self-organization of emergent phenomena and quintessentially underlying brain functions. Consequently, CFC works not only between the two ranges of slow harmonics of delta and theta frequencies, but also between central and peripheral rhythmic oscillations, thus providing resilience and robustness with homeostatic systems of individuals as a whole (S1 Fig).

## Break of CFC: A cease of harmonic oscillations followed by a loss of the coherence and power

Resilient CFC of slow harmonic waves seem to work throughout life, exhibiting stochastic vicissitude of delta or theta oscillations according to a responsible task. Cessation of bodily cardiorespiratory oscillations is considered as death of the individual. However, time-resolved

coherence between peripheral cardiorespiratory and brainstem delta-theta rhythms during and beyond the apparent cessation of oscillatory activity has not been described. Oscillation power is driven, to a large extent, by delta frequency waves, while theta waves seem to coordinate the delta-driven CFC power during a lifetime. In the present study, just before and during the cessation of apparent bodily cardiorespiratory oscillations, it should be noted that delta oscillations lost harmonic characteristics, a disappearance of the overtone components, while maintaining robust power for the time being (Figs 5 and 6). A break in CFC seemed to preclude maladaptive cooperation between delta and theta oscillations, resulting in a loss of power that can drive cardiorespiratory rhythms and CFC in the brainstem. The CFC of slow harmonics should be considered essential for maintaining life, that is, stable cardiorespiratory rhythms.

## The brainstem network with respect to whole-brain systems: A critical mesoscopic circuits

Brainstem neural networks are engaged in not only life-sustaining cardiorespiratory activity but also share varied functions such as cognition, sensorimotor, emotion, and interoception through connections with large-scale whole brain networks (S2 and S3 Figs). The brainstem not only provides relay stations for ascending sensory, descending motor, and other neural pathways involved with the various functions described above, but also houses concentrated neuronal cell groups of aminergic (catecholaminergic and serotonergic) and cholinergic neuronal systems controlling strategically large-scale brain functions through their brain-wide axonal projections, depending on the condition of individuals according to any incessant changes in the environment (S2 Fig). Thus, the brainstem seems to be critically involved in maintaining malleable whole-body homeostasis associated with fast sensorimotor functions and slow emotion or interoception, both of which could be interrelated with cardiorespiratory rhythmic activity. For this purpose, a communication strategy of the CFC of cardiorespiratory rhythms seems to be adopted primarily within the brainstem, based on the cooperative interplay between proposed distinctly separate but closely related oscillators [9, 22].

## Cardiorespiratory rhythmic system: Physiology

In the present study, peripheral cardiorespiratory rhythm coincided with the brain neuronal oscillatory activity. Fine-tuned reciprocal interactions between the peripheral and central oscillations based on concomitant cooperative slow harmonics and ranged delta-theta frequencies were clarified. Through the CFC mechanism of slow harmonics, it is likely that cooperative interactions between the brainstem and peripheral rhythms would be guaranteed.

The brainstem neural activity could represent a fluctuational cooperation generated by two types of distinct oscillator activities (somatic vs. visceral: respiratory vs. autonomic <cardiac>: 0–4 Hz delta vs. 4–10 Hz theta rhythms). The responsible brain regions of the coupling oscillators could be assigned to recurrent networks involving the NTS in the brainstem, especially a C1 neuronal group and the synaptically-connected neighboring areas [22] (S2 and S3 Figs). This fluctuation could represent an incessant change in a certain spatiotemporal dimension of neuronal activity synchrony by specific neuronal populations performing particular ongoing functions [8, 23]. In the present study, cooperative communications among certain groups of neuronal clusters through varied degrees of coupling via the two distinct oscillators were visualized and evaluated by wavelet analyses of the coherence and power of brain waves based on the FFT.

It has often been claimed that the rhythmicity recorded from the NTS could reflect only the peripheral oscillatory activities generated directly via the pulmonary and baro-/chemo-

receptor afferents (S1 Fig). However, this is likely not the case because the literature and our previous studies have shown that rhythmic activities in the brainstem are primarily of central origin. In short, they indicated the persistent presence of central rhythmicity reflecting cardiopulmonary activities even after a total deafferentation of vagotomy and sinoaortic denervation [7, 8, 14]. Moreover, the critical region of rhythmogenesis is localized in the rostral ventrolateral medulla (RVLM), which corresponds to the C1 and pre-Bötzinger regions [7, 14] (S2 and S3 Figs).

## Cardiorespiratory rhythmic system: Anatomy

Respiratory and cardiac rhythmic activities have been recorded in separate peripheral nerves and are thought to represent the respective signals controlling the thorax and heart musculature contraction (S1 Fig). It has been postulated that more robust rhythms of somatic respiratory musculature seem to be controlled phasically by fast-myelinated α-motoneuron nerves, while the sympathetic and parasympathetic signals of cardiac rhythmic activity are controlled by unmyelinated or thinly myelinated nerves, which are often referred to as tonic controls (S1 Fig). The efferent (motor) system of cardiorespiratory rhythmic activity originating in the brainstem is thought to be adjusted by the peripheral sensory apparatuses for baro-, chemoreception, and lung stretch (via the carotid sinus (IX), aortic (X), and vagus (X) nerves, respectively), which send the afferent information of change in the rhythm into the brainstem networks to adapt to a malleable environment (via unmyelinated afferents). Fast myelinated afferents originating from muscle spindles and Golgi tendon organs associated with thoracic respiration in skeletal muscles should participate in recurrent circuits. The efferent and afferent peripheral systems of cardiorespiratory rhythmic activity would converge in central brainstem neural networks. The vagal complex (VC) is one of the key neuronal populations that constitute brainstem neural networks and is strategically and critically involved in cardiorespiratory rhythm generation and maintenance (S3 Fig). The VC consists of the caudal nucleus of the tractus solitarius (cNTS), which serves as a sensory nucleus receiving visceral afferent information of any kind, including cardiorespiration, and the dorsal motor nucleus of the vagus nerve (dmnX), which represents a pool of parasympathetic motor efferent systems for the thoracic and abdominal viscera.

## Harmonics: Brain, body and daily life activities

Harmonics, wave oscillations with a unique fundamental frequency, and the following several overtones have rarely been described with respect to brain waves, because conventional electroencephalograms do not contain features of harmonic oscillations except conspicuous CFC without apparent harmonics [36]. In our present and previous studies [8, 9], it is claimed that brainstem wave oscillations exhibit a conspicuous feature of harmonics together with CFC and that the phenomenon is most saliently represented in the body as cardiorespiratory rhythmic activities. Further question arises as to why harmonic brain waves are recorded in the brainstem network, but not in the cerebral cortex. It should be noted that the main anatomical networks are composed of unmyelinated axons in the brainstem and myelinated axons in the telencephalon and diencephalon. We would like to stress on the anatomical difference in the solid network architecture because the degree of axon myelination could determine the conduction velocity of action potentials, thus exerting a critical effect on the oscillation frequency of wave activity.

Harmonics in our daily life are often described and reflected, particularly concerning music, electric transportation, and power grid systems. String musical instruments such as guitar, violin, and piano can produce harmonics that can often induce a deep impression on our

brains. The electric power transportation system handling a specific frequency of alternative electric waves must mitigate stochastic harmonics of the specific fundamental frequency that could cause tremendous mechanical damage to the power transportation system. It is likely that the system depends on the positive efficiency of transportation of electricity due to slower frequency harmonics generated spontaneously among the grid system.

## Conclusion

We propose that brain function may be based on self-organized emergent wave activity generated from incessant stochastic oscillations via functional networks temporarily available from anatomical connectivity, rather than on precise working of innumerable mechanical gears driven by pre-existing hard-wired anatomical networks. Emergent activity must be guaranteed from the point of view of reproducibility to such a degree that the activity contents can be communicated with other individuals, especially humans who use verbal language as the most efficient communication tool. I speculate that such fine-tuned brain activity must utilize a CFC mechanism that involves a higher frequency of brain wave activity, such as the gamma frequency. With a broad range of brain wave frequencies and resonant harmonics, the brain can manage various tasks of a wide spectrum of functions, such as communication using languages, from homeostasis maintenance to cognition. From an anatomical point of view, such a broad range of brain functions would be guaranteed by a hard-wired anatomical network consisting of unmyelinated and myelinated axons that could underlie slow harmonics and fine-tuned information, respectively. These functions should be based on a specific network configuration that is observed as a phylogenetically conserved framework: recurrent nested and coupled hard-wired networks.

## Supporting information

**S1 Fig. Peripheral cardiorespiratory and other oscillators: Networks connecting the periphery with the central nervous system (CNS).** The peripheral networks connected to the CNS involve various organs and muscles via both myelinated and unmyelinated nerves, blood-borne gases, electrolytes, and hormones. Voluntary muscles of respiration involve myelinated efferent and afferent nerves, while involuntary autonomic system both myelinated (preganglionic efferent) and unmyelinated (afferent and postganglionic efferent) peripheral nerves. Note the recurrent networks as oscillators for self-checking and accommodation. Dotted lines of ellipsoids show non-nerve transmission via gasses, electrodes and hormones. Abbreviations: A1-10, C1-3, catecholaminergic cell groups (A: dopaminergic or noradrenergic, C: adrenergic); B1-9, serotonergic cell groups; ACe, amygdaloid central nucleus; AP, area postrema; BNST, bed nucleus of the stria terminalis; Ch, Ch5-6, cholinergic cell groups; DLF, dorsal longitudinal fascicle; H, histaminergic cell group; HPV, hypothalamic paraventricular nucleus; ME, median eminence; MFB, medial forebrain bundle; NTS, nucleus of tractus solitarius; OVLT, organum vasculosum laminae terminalis; PAG, periaqueductal gray; PB, parabrachial nucleus; SFO, subfornical organ.
(TIF)

**S2 Fig. Ascending and descending networks involving the NTS and other structures associated with large-scale choline & amine projection systems and the circumventricular organs.** Sagittal and horizontal schematic brain drawings are shown with ascending (green) and descending (red) networks involving the NTS. Reciprocal networks extending from the telencephalon to the spinal cord involve cholinergic (Ch) and aminergic (A, B, C, H: noradrenergic, adrenergic, dopaminergic, serotonergic and histaminergic) neuronal clusters and

circumventricular organs (SFO, OVLT, ME, Neurohypophysis, AP), in addition to key nuclei such as PB, PAG, HPV, ACe, BNST via DLF and MFB. These structures may contribute to the maintenance of whole-body homeostasis via nervous and humoral bioregulation. **Abbreviations:** A1-10, C1-3, catecholaminergic cell groups (A: dopaminergic or noradrenergic, C: adrenergic); B1-9, serotonergic cell groups; ACe, amygdaloid central nucleus; AP, area postrema; BNST, bed nucleus of the stria terminalis; Ch, Ch5-6, cholinergic cell groups; DLF, dorsal longitudinal fascicle; H, histaminergic cell group; HPV, hypothalamic paraventricular nucleus; ME, median eminence; MFB, medial forebrain bundle; NTS, nucleus of tractus solitarius; OVLT, organum vasculosum laminae terminalis; PAG, periaqueductal gray; PB, parabrachial nucleus; SFO, subfornical organ.
(TIF)

**S3 Fig. Proposed brainstem oscillators as coupled and nested recurrent networks ((I) ~ (III)) possibly responsible for slow harmonics.** Sagittal and horizontal schematic drawings of the brainstem (pons and medulla oblongata) that contains presumed core oscillators generating delta and theta rhythms are presented. For details of the network configuration in the brainstem, especially the pons and medulla oblongata, see reference [22]. **Abbreviations:** A1-10, C1-3, catecholaminergic cell groups (A: dopaminergic or noradrenergic, C: adrenergic); B1-9, serotonergic cell groups; ACe, amygdaloid central nucleus; AP, area postrema; BNST, bed nucleus of the stria terminalis; Ch, Ch5-6, cholinergic cell groups; DLF, dorsal longitudinal fascicle; H, histaminergic cell group; HPV, hypothalamic paraventricular nucleus; ME, median eminence; MFB, medial forebrain bundle; NTS, nucleus of tractus solitarius; OVLT, organum vasculosum laminae terminalis; PAG, periaqueductal gray; PB, parabrachial nucleus; SFO, subfornical organ.
(TIF)

## Acknowledgments

The author would like to thank an anonymous native editor (Editage; www.editage.jp) for English language editing and valuable suggestions for the original manuscript.

## Author Contributions

**Conceptualization:** Yoshinori Kawai.

**Investigation:** Yoshinori Kawai.

**Writing – original draft:** Yoshinori Kawai.

**Writing – review & editing:** Yoshinori Kawai.

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
