## [Decision Letter · Decision Letter 0]

25 May 2023

PONE-D-23-04711Cross-Frequency Coupling between Slow Harmonics via the Real Brainstem Oscillators: An in vivo Animal StudyPLOS ONE

Dear Dr. Kawai,

Thank you for submitting your manuscript to PLOS ONE. After careful consideration, we feel that it has merit but does not fully meet PLOS ONE’s publication criteria as it currently stands. Therefore, we invite you to submit a revised version of the manuscript that addresses the points raised during the review process.

Please, revise introduction to emphasize the the lack of previous research in the area.

The sample size (number of rats) needs to be stated. Determining the statistical validity of the findings is difficult without this sample size stated in the materials and methods.

The reference section needs revision to ensure consistency in reference formatting.

Edit the figure legends for clarity and coherence with the main text (ex. lines 213-220).

Minor points

1. Abstract. Too long Abstract. First part, before "In this article …", includes sentences that belong to Introduction.

2. Line 127 and other epoch durations. Please,  provide the number of points of the epochs with the durations of 10, 20, and 100 s that were analyzed.

3. Line 167. What is the meaning of vertical bars in Panels Ba-Be of Fig. 2?

4. Line 182. 100 sec epoch is not longer than the previously mentioned 1000 sec epoch. Please correct wording. Please also introduce PZT, NTS, and LP in Fig. 3 in the first sentence, such as “Figure 3 shows … and so on”.

5. Line 188. I assume Figs. A1 and B1 stand for Fig. 3A1 and 3B1?

6. Line 190. There are no panels A2, A3, A4, B2, B3, and B4 in Fig. 2. Are those for Fig. 3?

7. Line 192. Should  "patterns in quality" be substituted with "qualitative patterns"

8. Line 194. What do arrows mean in panels A3 and A4 in Fig. 3?

9. Line 213. Standard deviation for Fig. 4Bb seems incorrect. It is of the order of 1.

10. Line 354. Here and further throughout the text: Fig. S1.

11. Lines 393-395. Please edit for clarity.

12. Line 400. The author can use either FFT or spell it out, but not both.

13. Lines 558-560. I am not clear about the meaning of the last sentence: It looks like coherence disappear, but panels do not show that signals disappear.

14. Line 613. Should you remove "two"?

15. Line 618. (d) and (q) must be removed.

We look forward to receiving your revised manuscript.

Kind regards,

Gennady S. Cymbalyuk, Ph.D.

Academic Editor

PLOS ONE

2. In order to comply with PLOS ONE's guidelines, please provide further details regarding housing conditions, feeding regimens, environmental enrichment, and all relevant steps taken to alleviate suffering (anesthesia, analgesia, details about humane endpoints, euthanasia, etc.). Please note that your Methods section should include sufficient information to be understood independently of any other methods or measures described elsewhere in your submission.

Reviewers' comments:

Reviewer's Responses to Questions

**Comments to the Author**

1. Is the manuscript technically sound, and do the data support the conclusions?

Reviewer #1: Partly

Reviewer #2: Yes

2. Has the statistical analysis been performed appropriately and rigorously? 

Reviewer #1: I Don't Know

Reviewer #2: Yes

3. Have the authors made all data underlying the findings in their manuscript fully available?

Reviewer #1: Yes

Reviewer #2: Yes

4. Is the manuscript presented in an intelligible fashion and written in standard English?

Reviewer #1: Yes

Reviewer #2: Yes

5. Review Comments to the Author

Reviewer #1: The introduction is too brief and relies heavily on the researcher’s own work. Additional references should be included to support the lack of previous research in the area. These could be reviews or articles that mention this gap in the literature.

The sample size (number of rats) needs to be stated. Determining the statistical validity of the findings is difficult without this sample size stated in the materials and methods.

The reference section needs revision due to inconsistent formatting. First, there are variations in the listing of the journal’s full name versus abbreviations (ex. 2 vs 4). Second, there are variations in the capitalization of the article name (ex. 4 vs 5). Third, the doi is not always listed even if it is available (ex. 25 vs. 26). The article should be edited to ensure consistency in reference formatting.

Several pieces of the paper could be included in the figure legends and vice versa (ex. lines 213-220).

Reviewer #2: In this paper, the authors investigated brainstem rhythms of the rat and their interactions. One of the rhythms seems to govern respiratory function (delta range) and another rhythm drives heart beats (theta range). It was demonstrated that these rhythms were synchronized in many cases of the brainstem activity by creation of temporary neural circuits. The author found a synchronization between central rhythmic and peripheral cardiorespiratory activities.

In general, the paper is well-written, the results seem solid, and the experimental data supports the conclusions of the paper. However, some sentences in the paper are not clearly written. Basically, writing style must be improved. However, I have only minor points that need to be corrected.

Minor points (lines are numbered as a sequence from the beginning of the paper)

1. Abstract. Too long Abstract. First part, before "In this article …", includes sentences that belong to Introduction.

2. Line 127 and other epoch durations. I would suggest to provide the number of points of the epochs with the durations of 10, 20, and 100 s that were analyzed. Those are important for the evaluation of the errors in Fast Fourier Transforms.

3. Line 167. What is the meaning of vertical bars in Panels Ba-Be of Fig. 2?

4. Line 182. 100 sec epoch is not longer than the previously mentioned 1000 sec epoch. Please correct wording. Please also introduce PZT, NTS, and LP in Fig. 3 in the first sentence, such as “Figure 3 shows … and so on”. I think, the reader must observe what is shown in the figure, which is quite complicated.

5. Line 188. I assume Figs. A1 and B1 stand for Fig. 3A1 and 3B1?

6. Line 190. There are no panels A2, A3, A4, B2, B3, and B4 in Fig. 2. I assume those are for Fig. 3?

7. Line 192. I think "patterns in quality" must be "qualitative patterns"

8. Line 194. What arrows mean in panels A3 and A4 in Fig. 3?

9. Line 213. Standard deviation for Fig. 4Bb seems incorrect. It is of the order of 1.

10. Line 354. Here and further throughout the text: Fig. S1.

11. Lines 393-395. Bad sentence. Please write more clear sentence or two.

12. Line 400. The author can use either FFT or spell it out, but not both. FFT has been defined above.

13. Lines 558-560. I am not clear about the meaning of the last sentence: It looks like coherence disappear, but panels do not show that signals disappear.

14. Line 613. I would remove "two".

15. Line 618. (d) and (q) must be removed.

6. PLOS authors have the option to publish the peer review history of their article (what does this mean?). If published, this will include your full peer review and any attached files.

Reviewer #1: No

Reviewer #2: No

---

## [Author Response · Author response to Decision Letter 0]

29 May 2023

29 May, 2023

Editorial Office 

PLOS ONE Journal

Dear Editorial Office:

I wish to submit a revised version of an Original Research Article (PONE-D-23-04711) for publication in PLOS ONE, titled “Cross-Frequency Coupling between Slow Harmonics via the Real Brainstem Oscillators: An in vivo Animal Study.” 

I appreciate the Reviewers’ careful and encouraging suggestions for revision of this article.

In that case, the following revisions or rebuttals (in red in below) were made as our response to comments from the Reviewers.

Reviewer #1: The introduction is too brief and relies heavily on the researcher’s own work. Additional references should be included to support the lack of previous research in the area. These could be reviews or articles that mention this gap in the literature.

This is a very important suggestion. Surprisingly, many physiologists of respiration have seemed to believe still that respiratory rhythmic activity is originated in specific neuronal groups in the brainstem as a pace-maker, not neuronal networks, while cardiac one is in the heart, in contrast to my claim that both rhythms of respiration and heart could be generated in the brainstem in a cooperative manner. Few of them have referred to any relationship with cardiac rhythm. Autonomic nervous system physiologists, on the other hand, have suggested that the cardiac rhythm may be associated with the brainstem neurons, however, few have mentioned any relationship with respiratory rhythm. I suspect that clinical cardiologists must recognize the interrelation of cardiorespiratory rhythm but could not yet describe this rhythm relation analytically. Therefore, as far as my knowledge adequate literature concerning cardiorespiratory cooperative activity in terms of rhythmic correlation could not be found for bridging the gap pointed out by this Reviewer. However, most recent references (5&6) concerning respiration networks were added newly with a sentence of “Furthermore, it has been generally stressed that cardiorespiratory rhythms be generated based on pacemaker-like activity originating from particular brainstem neurons, rather than network activity.” in the Introduction of the revised manuscript. (L.49-52)

According to a suggestion of “Please, revise introduction to emphasize the lack of previous research in the area” raised in the mail letter, sentences of “Furthermore, it has been generally stressed that cardiorespiratory rhythms be generated based on pacemaker-like activity originating from particular brainstem neurons, rather than network activity. However, the body-brain interrelationship of rhythmicity….” were incorporated with additional two references (5&6) in the revised manuscript. (LL.49-52)

The sample size (number of rats) needs to be stated. Determining the statistical validity of the findings is difficult without this sample size stated in the materials and methods.

According to the suggestions, number of rats were added in the revised version. (LL.98-99)

The reference section needs revision due to inconsistent formatting. First, there are variations in the listing of the journal’s full name versus abbreviations (ex. 2 vs 4). Second, there are variations in the capitalization of the article name (ex. 4 vs 5). Third, the doi is not always listed even if it is available (ex. 25 vs. 26). The article should be edited to ensure consistency in reference formatting.

The reference list was generated using EndNote with PubMed. However, as pointed out by the Reviewer (#1), the format of listing seems to be not consistent. I have reflected the consistency of the bibliography list notation in the revised version as far as I have noticed, as pointed out by this Reviewer. (LL.685-800)

Several pieces of the paper could be included in the figure legends and vice versa (ex. lines 213-220).

According to the suggestion, that part was changed. (LL.556-564)

Reviewer #2: In this paper, the authors investigated brainstem rhythms of the rat and their interactions. One of the rhythms seems to govern respiratory function (delta range) and another rhythm drives heart beats (theta range). It was demonstrated that these rhythms were synchronized in many cases of the brainstem activity by creation of temporary neural circuits. The author found a synchronization between central rhythmic and peripheral cardiorespiratory activities.

In general, the paper is well-written, the results seem solid, and the experimental data supports the conclusions of the paper. However, some sentences in the paper are not clearly written. Basically, writing style must be improved. However, I have only minor points that need to be corrected.

Minor points (lines are numbered as a sequence from the beginning of the paper)

1.Abstract. Too long Abstract. First part, before "In this article …", includes sentences that belong to Introduction.

According to the suggestion, a long sentence in the Abstract, “In this article, by focusing on the physiology and anatomy of a certain rodent brainstem region, the nucleus of the tractus solitarius, we demonstrated the characteristics of harmonic brain waves and their interactions, with a probable anatomical configuration of the responsible oscillator circuits.”, was moved to in the Introduction. (LL.82-85).

2. Line 127 and other epoch durations. I would suggest to provide the number of points of the epochs with the durations of 10, 20, and 100 s that were analyzed. Those are important for the evaluation of the errors in Fast Fourier Transforms.

According to the suggestion, a phrase of “with 1000 points” was added. (L.131)

3. Line 167. What is the meaning of vertical bars in Panels Ba-Be of Fig. 2?

A sentence of “Vertical bars in B indicate relative powers with respect to a common arbitrary unit.” was added in the Figure legends. (L.526) 

4. Line 182. 100 sec epoch is not longer than the previously mentioned 1000 sec epoch. Please correct wording. Please also introduce PZT, NTS, and LP in Fig. 3 in the first sentence, such as “Figure 3 shows … and so on”. I think, the reader must observe what is shown in the figure, which is quite complicated.

“A 1000 sec epoch” means a total recording duration of this experiment. From this long epoch recording, epochs of 10-20 sec or 100 sec were selected for FFT analyses. 

According to the suggestion, a sentence of “Figure 3 shows two examples of shorter epoch (~10 sec) recordings for coherence analyses between peripheral cardiorespiratory activities recorded by PZT and central neuronal MUAs in the NTS recorded by glass electrodes.” was added. (LL.187-189)

5. Line 188. I assume Figs. A1 and B1 stand for Fig. 3A1 and 3B1?

6. Line 190. There are no panels A2, A3, A4, B2, B3, and B4 in Fig. 2. I assume those are for Fig. 3?

Yes. The relevant figure numbers were carefully checked and all rewritten. (LL.195-209)

7. Line 192. I think "patterns in quality" must be "qualitative patterns"

According to the suggestion, the phrase was changed. (L.199)

8. Line 194. What arrows mean in panels A3 and A4 in Fig. 3?

Yes. The relevant figure numbers were carefully checked and all rewritten. (LL.195-209)

9. Line 213. Standard deviation for Fig. 4Bb seems incorrect. It is of the order of 1.

As the Reviewer pointed out, the digit in the values of the NTS potentials (mV) were wrongly described by one digit. The values were rewritten in the manuscript. The values in the Fig 4B seem to be correct. (LL.220-221)

10. Line 354. Here and further throughout the text: Fig. S1.

According to the suggestion, the words were changed throughout the text. :S# fig. � Fig. S#. S# and S## figs. � Figs. S# and S##.

11. Lines 393-395. Bad sentence. Please write more clear sentence or two.

The original sentence of “Brainstem neural activity was confirmed to reflect the fluctuational balance between two distinct oscillators (somatic and visceral: respiratory and autonomic <cardiac>: 0–4 Hz delta and 10 Hz theta).” was rewritten to “The brainstem neural activity could represent a fluctuational cooperation generated by two types of distinct oscillator activities (somatic vs. visceral: respiratory vs. autonomic <cardiac>: 0–4 Hz delta vs. 4–10 Hz theta rhythms).”. (LL.401-403)

12. Line 400. The author can use either FFT or spell it out, but not both. FFT has been defined above.

According to the suggestion, “the fast Fourier transformation (FFT)” was changed to “the FFT”. (L.408)

13. Lines 558-560. I am not clear about the meaning of the last sentence: It looks like coherence disappear, but panels do not show that signals disappear.

The original sentence of “After an apparent cessation of the peripheral and central oscillations, robust signals of delta range almost disappear with theta range signals as seen in (f) – (g).” was changed to “With minor peripheral and central activities of a noise level, robust signals of delta and theta range and their CFC coherence almost disappear as shown in (f) – (g).”. (LL.575-577)

14. Line 613. I would remove "two".

According to the suggestion, the word “two” was removed. (L.630)

15. Line 618. (d) and (q) must be removed.

(d) and (q) should read (�) and (�). (L.634-635)

==

I sent the revised version of the original manuscript with “Revised Manuscript with Track Changes” in which changes were shown in red.

Thank you for your consideration. I look forward to hearing from you.

Sincerely,

Yoshinori Kawai, PhD

Adati Institute for Brain Study (AIBS), 

Kawaguchi Saitama JAPAN, 333-0811

E-mail: ibs.stm.kwgt@gmail.com

---

## [Decision Letter · Decision Letter 1]

14 Jun 2023

PONE-D-23-04711R1Cross-Frequency Coupling between Slow Harmonics via the Real Brainstem Oscillators: An in vivo Animal StudyPLOS ONE

Dear Dr. Kawai,

Thank you for submitting your manuscript to PLOS ONE. After careful consideration, we feel that it has merit but does not fully meet PLOS ONE’s publication criteria as it currently stands. Therefore, we invite you to submit a revised version of the manuscript that addresses the points raised during the review process.

A few minor points are left to be addressed. Please, make sure that the abstract summarizes the most important results and their significance and does not exceed 300 words.

Line 199. The phrase "similar patterns in qualitative patterns" must be replaced by "similar qualitative patterns"

Figure 3. In the caption, the arrows in panels A3 and A4 should be described or removed.

Lines 401-403. Please clarify what brain regions create the two oscillators.

We look forward to receiving your revised manuscript.

Kind regards,

Gennady S. Cymbalyuk, Ph.D.

Academic Editor

PLOS ONE

Journal Requirements:

Reviewers' comments:

Reviewer's Responses to Questions

**Comments to the Author**

1. If the authors have adequately addressed your comments raised in a previous round of review and you feel that this manuscript is now acceptable for publication, you may indicate that here to bypass the “Comments to the Author” section, enter your conflict of interest statement in the “Confidential to Editor” section, and submit your "Accept" recommendation.

Reviewer #1: All comments have been addressed

Reviewer #2: (No Response)

2. Is the manuscript technically sound, and do the data support the conclusions?

Reviewer #1: (No Response)

Reviewer #2: Yes

3. Has the statistical analysis been performed appropriately and rigorously? 

Reviewer #1: (No Response)

Reviewer #2: Yes

4. Have the authors made all data underlying the findings in their manuscript fully available?

Reviewer #1: (No Response)

Reviewer #2: Yes

5. Is the manuscript presented in an intelligible fashion and written in standard English?

Reviewer #1: (No Response)

Reviewer #2: Yes

6. Review Comments to the Author

Reviewer #1: (No Response)

Reviewer #2: The author significantly improved the paper, but some comments were not interpreted correctly. I think the author still needs to make 4 minor corrections.

Minor points (lines are numbered as a sequence from the beginning of the paper)

1. Abstract. I suggested to shorten the first part of the Abstract, before the words "In this article …". However, the author moved to the Introduction the sentence started from "In this article …". As the results, Abstract does not describe what is done in the paper. Please shorten the first part of the Abstract from the original submission.

I am providing Abstract requirements for PLoS ONE:

“The Abstract should:

• Describe the main objective(s) of the study

• Explain how the study was done, including any model organisms used, without methodological detail

• Summarize the most important results and their significance

• Not exceed 300 words”

2. Line 199. Phrase "similar patterns in qualitative patterns" must be replaced by "similar qualitative patterns"

3. Figure 3. What arrows mean in panels A3 and A4 in Fig. 3? This question still was not answered. The author needs to remove arrows from the figure, if they are not necessary.

4. Lines 401-403. Still bad sentence. Please explain more clearly what brain regions create two oscillators. It is not clear as it is written in the parentheses.

7. PLOS authors have the option to publish the peer review history of their article (what does this mean?). If published, this will include your full peer review and any attached files.

Reviewer #1: No

Reviewer #2: No

---

## [Author Response · Author response to Decision Letter 1]

17 Jun 2023

18 June, 2023

Editorial Office 

PLOS ONE Journal

Dear Editorial Office:

I wish to submit a revised version of an Original Research Article (ONE-D-23-04711R1) for publication in PLOS ONE, titled “Cross-Frequency Coupling between Slow Harmonics via the Real Brainstem Oscillators: An in vivo Animal Study.” 

I appreciate the Reviewers’ careful and encouraging suggestions for revision of this article.

In that case, the following revisions or rebuttals (in red in below) were made as our response to comments from the Reviewer.

Please, make sure that the abstract summarizes the most important results and their significance and does not exceed 300 words.

1. Abstract. I suggested to shorten the first part of the Abstract, before the words "In this article …". However, the author moved to the Introduction the sentence started from "In this article …". As the results, Abstract does not describe what is done in the paper. Please shorten the first part of the Abstract from the original submission.

According to the suggestion, a sentence of “The recorded brain waves represent oscillatory synchrony of electrical activity generated by the varied spatial sizes of neuronal assemblies that are activated simultaneously.” was removed. As indicated in red in the revised manuscript, several words were added to imply an animal model and comply with other recommendations.

 As the results, I believe that the Abstract summarizes the most important results and their significance more succinctly than before, and confirmed that the abstract does not exceed 300 words (177 words).

2. Line 199. The phrase "similar patterns in qualitative patterns" must be replaced by "similar qualitative patterns"

According to the suggestion, the phrase "similar patterns in qualitative patterns" was replaced by "similar qualitative patterns". (L.199)

3. Figure 3. In the caption, the arrows in panels A3 and A4 should be described or removed.

According to the suggestion, the phrase “indicated by arrows” was added in the figure legend. (L.545)

4. Lines 401-403. Please clarify what brain regions create the two oscillators.

According to the suggestion, a sentence of “The responsible brain regions of the coupling oscillators could be assigned to recurrent networks involving the NTS in the brainstem, especially a C1 neuronal group and the synaptically-connected neighboring areas [22] (Figs. S2 and S3).” was incorporated. (LL.404-406)

==

I sent the revised version of the original manuscript with “Revised Manuscript with Track Changes” in which changes were shown in red.

Thank you for your consideration. I look forward to hearing from you.

Sincerely,

Yoshinori Kawai, PhD

Adati Institute for Brain Study (AIBS), 

Kawaguchi Saitama JAPAN, 333-0811

E-mail: ibs.stm.kwgt@gmail.com

---

## [Decision Letter · Decision Letter 2]

24 Jul 2023

Cross-Frequency Coupling between Slow Harmonics via the Real Brainstem Oscillators: An in vivo Animal Study

PONE-D-23-04711R2

Dear Dr. Kawai,

We’re pleased to inform you that your manuscript has been judged scientifically suitable for publication and will be formally accepted for publication once it meets all outstanding technical requirements.

Kind regards,

Gennady S. Cymbalyuk, Ph.D.

Academic Editor

PLOS ONE

Additional Editor Comments (optional):

Reviewers' comments:

Reviewer's Responses to Questions

**Comments to the Author**

1. If the authors have adequately addressed your comments raised in a previous round of review and you feel that this manuscript is now acceptable for publication, you may indicate that here to bypass the “Comments to the Author” section, enter your conflict of interest statement in the “Confidential to Editor” section, and submit your "Accept" recommendation.

Reviewer #2: All comments have been addressed

2. Is the manuscript technically sound, and do the data support the conclusions?

Reviewer #2: Yes

3. Has the statistical analysis been performed appropriately and rigorously? 

Reviewer #2: Yes

4. Have the authors made all data underlying the findings in their manuscript fully available?

Reviewer #2: Yes

5. Is the manuscript presented in an intelligible fashion and written in standard English?

Reviewer #2: Yes

6. Review Comments to the Author

Reviewer #2: All my comments have been properly addressed. I have no further concerns or comments. Best regards.

7. PLOS authors have the option to publish the peer review history of their article (what does this mean?). If published, this will include your full peer review and any attached files.

Reviewer #2: No

---

## [Editor Report · Acceptance letter]

27 Jul 2023

PONE-D-23-04711R2 

Cross-Frequency Coupling between Slow Harmonics via the Real Brainstem Oscillators: An *in vivo* Animal Study 

Dear Dr. Kawai:

I'm pleased to inform you that your manuscript has been deemed suitable for publication in PLOS ONE. Congratulations! Your manuscript is now with our production department. 

Kind regards, 

on behalf of

Dr. Gennady S. Cymbalyuk 

Academic Editor

PLOS ONE